# Prostate Cancer Brain Metastasis: Review of a Rare Complication with Limited Treatment Options and Poor Prognosis

**DOI:** 10.3390/jcm11144165

**Published:** 2022-07-18

**Authors:** Kobisha Rajeswaran, Kaitlin Muzio, Juan Briones, Mary Jane Lim-Fat, Chia-Lin Tseng, Martin Smoragiewicz, Jay Detsky, Urban Emmenegger

**Affiliations:** 1Division of Medical Oncology, Odette Cancer Centre, Sunnybrook Health Sciences Centre, University of Toronto, 2075 Bayview Avenue, Toronto, ON M4N 3M5, Canada; kobisha228@gmail.com (K.R.); kaitlin.muzio@gmail.com (K.M.); martin.smoragiewicz@sunnybrook.ca (M.S.); 2Department of Hematology-Oncology, School of Medicine, Pontificia Universidad Católica de Chile, 309 Diagonal Paraguay, Santiago 8330077, Chile; jbrionesc@gmail.com; 3Division of Neurology, Department of Medicine, Sunnybrook Health Sciences Centre, University of Toronto, 2075 Bayview Avenue, Toronto, ON M4N 3M5, Canada; maryjane.limfat@sunnybrook.ca; 4Department of Radiation Oncology, Odette Cancer Centre, Sunnybrook Health Sciences Centre, University of Toronto, 2075 Bayview Avenue, Toronto, ON M4N 3M5, Canada; chia-lin.tseng@sunnybrook.ca (C.-L.T.); jay.detsky@sunnybrook.ca (J.D.); 5Department of Medicine, University of Toronto, 1 King’s College Cir, Toronto, ON M5S 1A8, Canada; 6Biological Sciences Research Platform, Sunnybrook Research Institute, Sunnybrook Health Sciences Centre, University of Toronto, 2075 Bayview Avenue, Toronto, ON M4N 3M5, Canada; 7Institute of Medical Science, University of Toronto, 1 King’s College Cir, Toronto, ON M5S 1A8, Canada

**Keywords:** prostatic neoplasms, brain metastasis, systematic literature review, treatment, outcome

## Abstract

Brain metastases (BM) are perceived as a rare complication of prostate cancer associated with poor outcome. Due to limited published data, we conducted a literature review regarding incidence, clinical characteristics, treatment options, and outcomes of patients with prostate cancer BM. A literature analysis of the PubMed, MEDLINE, and EMBASE databases was performed for full-text published articles on patients diagnosed with BM from prostate cancer. Eligible studies included four or more patients. Twenty-seven publications were selected and analyzed. The sources of published patient cohorts were retrospective chart reviews, administrative healthcare databases, autopsy records, and case series. BM are rare, with an incidence of 1.14% across publications that mainly focus on intraparenchymal metastases. Synchronous visceral metastasis and rare histological prostate cancer subtypes are associated with an increased rate of BM. Many patients do not receive brain metastasis-directed local therapy and the median survival after BM diagnosis is poor, notably in patients with multiple BM, dural-based metastases, or leptomeningeal dissemination. Overall, prostate cancer BM are rare and associated with poor prognosis. Future research is needed to study the impact of novel prostate cancer therapeutics on BM incidence, to identify patients at risk of BM, and to characterize molecular treatment targets.

## 1. Introduction

Prostate cancer is the second most common cancer in men globally, accounting for almost 1.4 million new cases and around 375,000 deaths in 2020 [1]. The clinical presentation of prostate cancer can range from an asymptomatic, low-risk localized form to aggressive metastatic disease. For the approximately 17% of prostate cancer patients who develop metastases, the most common metastatic locations are the skeleton (84.4%), distant lymph nodes (10.6%), liver (10.2%), and lung (9.1%) [2,3].

One of the rarer locations of metastasis from prostate cancer is the brain, with an estimated incidence of less than 2% [2,4,5]. Central nervous system (CNS) dissemination includes parenchymal and dural-based (pachymeningeal) brain metastases, and leptomeningeal disease [6]. Although considered a rare metastatic site, there has been a perceived increase in the incidence of prostate cancer brain metastases (BM) over time [7]. This rise is attributed to longer survival due to improved systemic therapy, allowing sufficient time for metastases to progress to the brain [8], and more advanced diagnostic techniques, enabling earlier detection of BM [9,10].

Amongst all metastatic sites from prostate cancer, BM are associated with particularly poor outcome, with the most recent available patient series suggesting a median survival ranging from two to twelve months following diagnosis [4,11,12]. Furthermore, BM from prostate cancer are also associated with a high symptom burden [13,14]. 

Patients with BM usually are excluded from clinical trials and there are no established systemic or intrathecal treatment options for prostate cancer BM. Docetaxel chemotherapy cannot easily penetrate the blood–brain barrier due to exclusion by drug efflux pumps and is not suitable for intrathecal administration [15]. Cabazitaxel chemotherapy more easily passes the blood–brain barrier in rodent models compared to docetaxel but it remains unclear whether this results in clinically meaningful intracerebral anti-prostate cancer activity [16]. Second-generation androgen receptor-signaling inhibitors are either largely excluded from the CNS (e.g., darolutamide) or achieve intracerebral concentrations associated with a small risk of seizures (e.g., apalutamide and enzalutamide). This seizure risk is thought to be amplified by BM amongst other risk factors [17,18,19]. The androgen synthesis inhibitor abiraterone passes the blood–brain barrier, yet there is a lack of evidence on whether this results in clinically significant antineoplastic activity [20]. While there are a few exceptions (e.g., niraparib), generally poly (ADP-ribose) polymerase inhibitors (including olaparib, rucaparib, and talazoparib) pass the blood–brain barrier only poorly [21,22,23,24,25,26]. Finally, radium-223 is a bone-targeted radioisotope without soft tissue anti-cancer activity [27], while anecdotal responses of prostate cancer BM to 177Lu-PSMA-617 await confirmation in clinical studies [28,29].

Due to the rarity of prostate cancer BM, there is a lack of comprehensive clinicopathological information and evidence-based consensus regarding patient management strategies. Hence, we sought to review available studies related to prostate cancer BM in an effort to better understand the incidence, treatment options, and outcome of this rare complication.

## 2. Materials and Methods

### 2.1. Aim and Design

The purpose of the literature search was to identify available articles regarding BM from prostate cancer, guided by the main research question: what is known from the literature on the occurrence, treatment, and outcome of patients with BM from prostate cancer?

### 2.2. Search Strategy

The PubMed, MEDLINE, and EMBASE databases were searched from inception to 14 September 2021, for fully published articles using the following search string: prostatic neoplasms AND (dural OR leptomeningeal OR intracranial OR parenchymal OR “brain metastasis” OR “brain metastases”). The search results were independently reviewed by two reviewers (KM and KR) and publications were included according to the eligibility criteria outlined below. Additionally, a manual reference search of the articles selected from the database search was conducted to include any other relevant and applicable publications that may have been missed by the key-term search.

### 2.3. Eligibility Criteria

The present analysis included articles focusing on BM from prostate cancer; additional articles were considered if a sufficient sub-cohort of patients with BM from prostate cancer was identified and separately described. Prospective studies, retrospective studies, and case series were all acceptable study designs. We included studies with at least four patients and that were written in English. Conference abstracts, review articles, and editorials were excluded.

### 2.4. Data Extraction and Analysis

Using a predefined data extraction template, two reviewers (KM and KR) independently extracted information from the selected articles, including the number of patients, BM incidence rates, baseline prostate cancer characteristics and patient demographics, prognostic factors for BM, the types and characteristics of metastases studied, and information on prior prostate cancer treatments as well as the treatment of BM. Any discrepancies were resolved by consensus or consultation with a third reviewer (UE). To analyze patient characteristics, treatment patterns, and outcome measures, various descriptive statistics, such as percentages, medians, and ranges, were used. Time to event outcome measures were studied using Kaplan–Meier analysis (GraphPad Prism 9, GraphPad Software, San Diego, CA, USA).

## 3. Results

### 3.1. Literature Search

Figure 1 depicts the flow diagram for the literature search. In total, 723 publications were identified. Following removal of duplicates and after application of the predefined eligibility criteria, we identified 23 articles. The manual reference search conducted for each of the 23 selected articles resulted in the selection of an additional four articles. Ultimately, a total of 27 articles were included in this analysis, spanning the period from 1976 to 2021 [4,5,7,9,10,11,12,13,14,30,31,32,33,34,35,36,37,38,39,40,41,42,43,44,45,46,47]. 

### 3.2. Study Characteristics

The majority of identified articles (*n* = 17; 63%) relied on patient data collected from the United States (Figure 2A). The next most represented countries by number of publications are Italy, Germany, and Australia, accounting for four (15%), two (7%), and two (7%) articles, respectively. Most of the publications included in this review are from 2001 to 2021, representing 20 of the 27 (74%) selected articles (Figure 2B). As noted in Table 1 and depicted in Figure 2C, the median number of patients per study was 13, with a range from 4 (the lower limit as per predefined selection criteria; case series and autopsy reviews) to 187 (administrative healthcare database study). Patient data was reported from four possible sources: retrospective chart reviews (*n* = 13; 48%), administrative healthcare databases (*n* = 7; 26%), autopsy series (*n* = 4; 15%), and case series (*n* = 3; 11%; Table 1; Figure 2C). 

### 3.3. Disease Characteristics

Six studies (22.2%) described patients with castration-resistant prostate cancer (CRPC) at the time of diagnosis of BM while two administrative database studies (7.4%) focused on castration-sensitive prostate cancer (CSPC). Nine studies (33.3%) had mixed patient populations and in ten studies (37%), no corresponding specifications were provided (Table 1). The proportion of patients with de novo versus metachronous BM was not reported in many of the studies. Of the studies that did report this information, the majority of patients had metachronous BM [7,38,47], whereas Ganau et al. analyzed a cohort of patients with a plurality of de novo BM (10/19; 53%) [40]. The incidence of BM was reported in 14 of 27 studies (52%), with a median incidence across these informative studies calculated as 1.14% (range of 0.16–8.06%; Table 1). Worthy of note, the incidence rates reported by Bhambhvani et al., Caffo et al., and Kanyılmaz et al. were not included in this calculation; these three publications first searched records of patients with BM of diverse origins and then identified the percentages of cases that were deemed to be caused by prostate cancer (0.86%, 1.84%, and 2.95%, respectively) [7,11,13]. Hatzoglou et al. reported a higher incidence of BM in patients with rarer histological subtypes of prostate cancer when compared to adenocarcinoma (0.13%; 18/13,547), with 10% (1/10) of small cell carcinoma patients and 25% (1/4) of patients with neuroendocrine prostate cancer [38]. The median time from diagnosis of prostate cancer to diagnosis of BM was 42 months, with a range of 6–90 months (Table 1).

Fourteen articles (52%) reported exclusively on parenchymal BM, whereas two studies (7%) involved only patients with metastases to the dura, one study (4%) focused on leptomeningeal metastases, and ten analyses (37%) comprised mixed populations (Table 2). The median number of BM reported across all the studies was one (range of 1–8), whereas the median percentage of patients presenting with single or multiple BM was 75% and 25%, respectively. Hemorrhagic BM were rarely documented, although it was noted in Hatzoglou et al. that 33% of patients had at least one hemorrhagic lesion and in Taylor et al., 64% of patients had either adjacent brain compression or hemorrhage [33,38].

If reported (19/27 studies; 70%), the most common location of concurrent extracerebral metastases was to the bones, with a median of 95% of patients across articles presenting with bone metastases (Table 2). In three studies, the locations of synchronous metastases were not reported by specific location but instead reported as visceral metastases. Each of those particular studies reported high rates of concurrent visceral metastases ranging from 59% to 81% [12,37,40].

### 3.4. Prior Treatment History and Prostate Cancer Brain Metastases Management

Whenever reported, data was collected on the type of prostate cancer therapy received prior to the diagnosis of BM. Across the studies, the most reported treatments were radiation therapy to the prostate (*n* = 9; 33%), prostatectomy (*n* = 8; 30%), androgen deprivation therapy (*n* = 8; 30%), and unspecified chemotherapy (*n* = 8; 30%). Worthy of note, only one study reported on the use of enzalutamide and abiraterone, with 80% of patients from the study by Boxley et al. with a history of abiraterone or enzalutamide exposure prior to BM development [5]. There was insufficient information on the number of lines of therapy that patients had received for prostate cancer before their BM diagnosis.

The majority of the reviewed articles (*n* = 19; 70%) also reported on BM treatment (Table 3). The most common therapies specified among the studies were radiation (*n* = 15; 55.6%), surgery and radiation (*n* = 7; 26%), supportive care (*n* = 7; 26%), and surgery only (*n* = 5; 19%). Of the sixteen studies that reported details with respect to the type of radiation therapy applied, nine (56%) used whole-brain radiotherapy only, three (19%) used stereotactic radiosurgery, and four (25%) combined both modalities. Limited information was presented that differentiated the patient characteristics leading to receiving WBRT versus SRS. Additionally, 22% (6/27) of studies reported the use of systemic chemotherapy following BM diagnosis, with docetaxel, mitoxantrone, and cyclophosphamide being the only specified treatments used [4,10,34,44,45,47].

### 3.5. Outcome and Prognostic Factors

Across informative studies, the median overall survival from diagnosis of BM was 4.5 months, ranging from 1 to 13 months (Table 3). Two studies reported survival as a function of the type of treatment received. In the analysis by Tremont-Lukats et al., the median survival was one month for untreated patients versus three months for patients treated with radiation therapy [14]. Bhambhvani et al. found a median survival of 1.2 months for untreated patients, 4.6 months for patients treated with RT, and 13 months for patients with combined resection plus radiation therapy [11].

Commonly, patients with intraparenchymal, dural-based, and leptomeningeal metastases were lumped together for outcome analyses, but a few studies detailed patient and disease characteristics as well as outcomes by sub-cohorts. Boxley et al. split the reporting of the Gleason score, interval between prostate cancer and BM diagnosis, and overall survival from diagnosis of BM by intraparenchymal versus dural-based metastasis sub-populations [5]. For patients with intraparenchymal metastases, the median Gleason score was eight and the median time from prostate cancer diagnosis to diagnosis of BM was 8.36 years, compared to a median Gleason score of nine and a median time from prostate cancer diagnosis to BM diagnosis of 3.62 years in patients with dural-based metastases. The median overall survival was longer for patients with intraparenchymal metastases (5.4 months) than for those with dural-based metastases (2.6 months). Similarly, Caffo et al. recorded the median survival for dural versus leptomeningeal metastasis sub-populations, with a median of four months and one month, respectively [47]. Additionally, there was one article that studied four patients exclusively with leptomeningeal metastases; however, no analysis of survival was reported within this study [10]. Tremont-Lukats et al. calculated the median survival after a diagnosis of BM by the histological type of prostate cancer. The median survival was shorter for patients with adenocarcinoma (one month) than for patients with other histologic subtypes, such as small cell carcinoma, squamous cell tumors, and rhabdomyosarcoma (six months), although this difference was not found to be statistically significant [14]. 

Prognostic factors affecting survival of prostate cancer patients with BM were highlighted in several articles. Treatment administration was positively associated with increased survival outcomes [10,11,31,32,34,40,44]. In one study of 31 patients, radiation alone and surgical resection combined with radiation produced hazard ratios of 0.11 and 0.05 for survival, respectively, in comparison to patients who received no treatment [11]. Intraparenchymal metastases (compared to dural or leptomeningeal metastases) and single cerebral lesions (relative risk of 1.54 for multiple metastases) were also associated with improved overall survival [5,30,36,40]. The presence of extracerebral metastases other than bone was another unfavorable prognostic factor; in particular, several studies note that patients with liver metastases had poor survival [4,30,44]. Other factors negatively affecting overall survival included a poor Karnofsky Performance Score (KPS; hazard ratio of 3.18 for survival in patients with KPS ≤ 70 versus ≥80), high prostate-specific antigen (PSA) levels, and histological types of prostate cancer other than classical adenocarcinoma [12,14,31,33,39,40].

### 3.6. Patient and Disease Characteristics Associated with Brain Metastasis

Myint and Qasrawi reported an increased risk for BM in patients with CSPC and concurrent visceral metastases, with multivariate odds ratios for liver and lung metastasis of 2.85 (95% CI; 1.89–4.2) and 4.6 (95% CI; 3.3–6.4), respectively [4]. Other predictive factors described for prostate cancer BM included non-adenocarcinoma and rare histological types of prostate cancer, such as small-cell carcinoma [11,14,31,34,38]. However, these latter studies lacked definite statistical confirmation of the suggested associations.

## 4. Discussion

The present literature review of 27 articles on prostate cancer BM imparts several key findings. First, clinically diagnosed prostate cancer BM are rare (<2%), even in contemporary publications. Autopsy series from the 1970s and 1980s describe higher incidence rates in the range of 4%, which suggests that some prostate cancer BM may remain undetected despite significant improvements of imaging techniques. Worthy of note, the standard of care staging of patients with advanced prostate cancer does not involve CNS imaging in the absence of suspicious symptoms [48]. Due to inter-study heterogeneity, we were not able to test the hypothesis that the incidence of BM from prostate cancer has been increasing recently because of improved imaging methods and more extensive treatment exposure [47]. With respect to the latter, the two large administrative database analyses describe a similar incidence of BM in treatment-naïve patients (0.76% and 1.26%) compared to the median rate calculated across our analysis (1.14%) [4,44]. On the other hand, Lawton et al. reported the highest incidence of prostate cancer BM (10/124; 8%) within a highly selected cohort of young patients (median age of 59 years) with exclusively dural-based metastases from CRPC and a median Gleason score of nine at initial diagnosis [45].

Secondly, in a descending order of reported frequency, prostate cancer BM are intraparenchymal, dural-based (pachymeningeal), or leptomeningeal. Most studies focused on intraparenchymal metastases, which appear to be associated with improved survival compared to dural-based or leptomeningeal metastases [5,47]. This is interesting to note considering dural-based metastases have been associated with better survival in other cancers [49,50]. However, the paucity of published information and possible publication bias preclude definite conclusions regarding the impact of the metastatic location within the CNS on patient outcome.

Thirdly, bone was the most common location for synchronous extracerebral metastases, as would be expected given the presence of bone metastases in around 90% of patients with metastatic prostate cancer [51]. On the other hand, concurrent liver and lung metastases seem overrepresented at a rate of around 30% [52]. In fact, in their series of patients with de novo prostate cancer BM, Myint and Qasrawi identified the synchronous presence of liver or lung metastases as independent risk factors for BM [4]. Additionally, histologic types other than adenocarcinoma (i.e., small cell carcinoma, primary transitional cell carcinoma, and neuroendocrine prostate cancer) were all noted as possible predictive factors for BM [11,14,31,33,34,38,39]. Notably, only one article included in our analysis by Ormond et al. presented information on molecular alterations found in BM (i.e., ERG, CHD1, PTEN, and MAP3K7 immunohistochemistry) [46]. From our understanding, this is one of only four publications in existing literature that attempts to study molecular changes in regard to prostate cancer BM using patient samples [46,53,54,55]. Nguyen et al. described an enrichment for androgen receptor amplification and NOTCH pathway aberrations in prostate cancer BM [55]. While the former was also found in bone metastases, NOTCH aberrations were a unique feature of prostate cancer BM compared to other metastatic sites such as bone, liver, and lung. Otherwise, it remains to be seen whether prostate cancer BM harbor distinct actionable molecular targets compared to primary tumors or extracerebral metastases.

Fourthly, the median survival of patients with prostate cancer BM is poor, at 4.5 months across studies, although surgical resection and/or radiation therapy may achieve longer-term local disease control, notably in patients with solitary BM [42]. Rare studies describe anecdotal cases of patients living for more than one year after BM diagnosis, including one patient surviving over 22 months, as reported by Gzell et al. and Caffo et al. [7,39]. Worthy of note, these two patients were the youngest of their cohorts, with one having de novo BM [39]. Due to low CNS-penetration (i.e., docetaxel) and other shortcomings (i.e., possible seizure induction by enzalutamide or apalutamide), there is no defined role for systemic therapy in the management of prostate cancer BM to date. However, changing systemic prostate cancer therapies at the time of BM diagnosis may improve the survival of patients by controlling extracerebral prostate cancer manifestations as a competing cause of death as long as the BM are controlled by local means such as radiation therapy and/or resection. In fact, metastases-directed therapy or the use of chemotherapy was found to be associated with improved overall survival [4,11]. Nonetheless, a sizeable number of patients do not receive any type of metastases-directed or further systemic therapy at diagnosis of BM, including in contemporary series. 

Our study also reveals numerous limitations of the available evidence. Due to the rare occurrence of BM from prostate cancer, there were considerably more case reports available (115/374 total articles; 31%) than comprehensive studies. The selected articles span a period of 45 years that saw major changes in the management of advanced prostate cancer and diagnostic options, including regarding the radiological detection of BM. Most studies originate from the United States and Europe, whereas there is a lack of information on prostate cancer BM in other geographical areas. Many of the reports focus on intraparenchymal BM, yet considerably less is known regarding dural-based or leptomeningeal metastases. Due to sporadic reporting practices, we did not include biochemical parameters (such as PSA) in our analysis. Due to the rarity of surgical resection of prostate cancer BM, histopathological confirmation of the prostate cancer origin of CNS lesions is rare. Low resection rates contribute to the lack of comprehensive information on the role of prostate cancer variants (e.g., small cell or neuroendocrine prostate cancer) as mediators of BM and the paucity of information regarding possible molecular drivers and therapeutic targets of prostate cancer metastases to the brain, particularly in comparison to other malignancies [46,53,54,55,56]. Aside from the study by Boxley et al., there is a dearth of information on prostate cancer BM in patients with prior exposure to second-generation androgen receptor-signaling inhibitors, such as abiraterone or enzalutamide, which have become the most used group of agents for the treatment of advanced prostate cancer over the last decade [5]. To date, there is no prospective evidence with respect to patient and/or disease characteristics predicting an increased risk of prostate cancer BM, which could be used for guidance regarding BM screening in patients with prostate cancer.

In summary, while BM from prostate cancer are both rare, yet possibly underdiagnosed, the associated prognosis is poor, with no apparent recent improvement in survival. A sizeable proportion of patients do not receive metastases-directed therapy. Looking ahead, our analysis identified several areas of future research, including the impact of novel prostate cancer therapeutics on the incidence of BM, the molecular characterization of prostate cancer BM, and the clinical exploration of systemic treatment options suitable for complementing surgical resection and/or BM-targeted radiation therapy. With respect to the latter, it is worth noting that not only are prostate cancer patients with BM typically excluded from clinical trial participation but, to the best of our knowledge, there are also no PCBM-specific clinical studies exploring systemic treatment options (https://www.clinicaltrials.gov, accessed on 30 June 2022). Efforts are needed to develop tools that predict the presence of prostate cancer BM, which would enable the identification of BM at an early stage when more patients would be able to tolerate local therapy or enrollment in trials exploring novel treatment strategies.

## Figures and Tables

**Figure 1 jcm-11-04165-f001:**
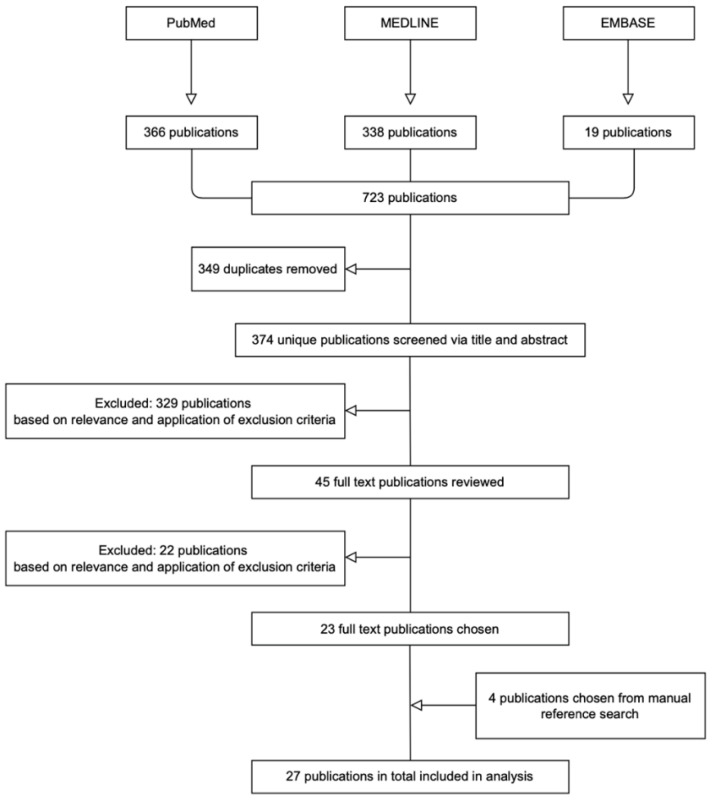
Flow diagram of selected publications. PubMed, MEDLINE, and EMBASE databases were searched. Publications were filtered by title, abstract, and exclusion criteria.

**Figure 2 jcm-11-04165-f002:**
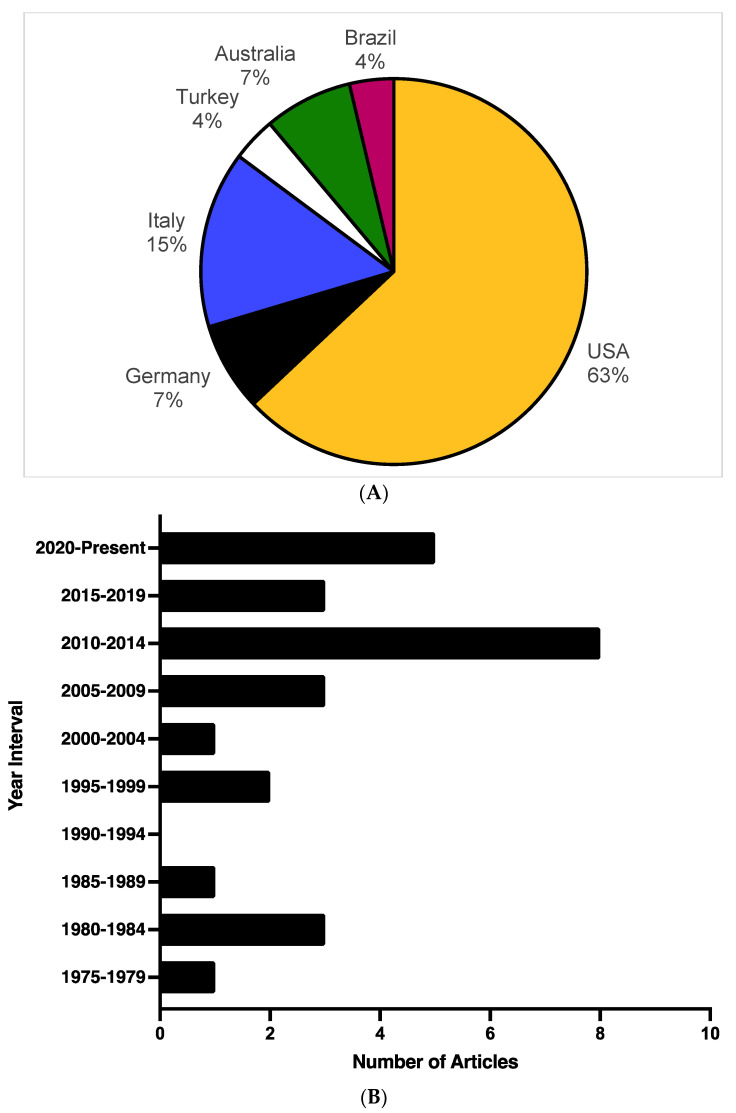
(**A**–**C**) Publication characteristics: the selected articles were analyzed to understand the characteristics regarding the country of study, temporal publication trends, as well as the number of patients per study and source of patient information. (**A**) Publication by country of study: number of publications by country of study. (**B**) Publication by year: depiction of the number of publications per five-year intervals. (**C**) Publication by patient selection: number of patients per study and assignment of publications to the following sources of information: autopsy reviews, case series, retrospective chart reviews, or administrative database search [4,5,7,9,10,11,12,13,14,30,31,32,33,34,35,36,37,38,39,40,41,42,43,44,45,46,47].

**Table 1 jcm-11-04165-t001:** Study and patient characteristics.

Reference	N = Total Number of Patients	Patient Selection Method	% Incidence Rate of Brain Metastases	Median Time between Diagnoses in Months (range) ^a^	Prostate Cancer Type ^b^		
					CSPC	CRPC	Not Specified
Myint ZW 2021 [4]	187	AD	1.26		187		
Boxley PJ 2021 [5]	29	AD	0.44		4	25	
Bhambhvani HP 2020 [11]	31	RC		81 (3–195)			31
Nguyen T 2020 [12]	21	RC					21
Ganau M 2020 [40]	19	RC			10	9	
Ormond DR 2019 [46]	21	AD		90 (0–312)	1	20	
Zhao F 2019 [44]	126	AD	0.76		126		
Kanyılmaz G 2019 [13]	10	RC					10
Hatzoglou V 2014 [38]	21	RC	0.16	46			21
Bartscht T 2014 [30]	18	RC					18
Gzell CE 2013 [39]	4	CS			1	3	
Caffo O 2013 [7]	9	AD		36 (0–111)		9	
Caffo O 2012 [47]	31	AD	3.29	44 (6–173)		31	
Guedes B 2011 [37]	6	CS			3	3	
Lawton A 2010 [45]	10	RC	8.06	40 (21–164)		10	
Flannery T 2010 [31]	10	RC	1.01	36 (12–180)	1	9	
Lin C 2008 [10]	4	CS				4	
Kim SH 2008 [32]	5	AD		82		5	
Salvati M 2005 [9]	13	RC		45 (mean)			13
Tremont-Lukats IW 2003 [14]	103	RC	0.63				103
McCutcheon IE 1999 [34]	38	RC	0.48	29 (0–84)			38
Nussbaum ER 1996 [36]	11	RC		22			11
Chung TS 1986 [42]	8	RC	0.61	6 (0–73)	2	6	
Taylor HG 1984 [33]	14	AR	4.13		5	9	
Castaldo JE 1983 [43]	8	AR	4.23				8
Sarma DP 1983 [35]	4	AR	3.31		3	1	
Catane R 1976 [41]	5	AR	4.40	61 (mean)		5	
Overall Median	13	N/A	1.14	42	N/A	N/A	N/A

AD: administrative database review; RC: retrospective chart review; CS: case series; AR: autopsy review; CSPC: castration-sensitive prostate cancer; CRPC: castration-resistant prostate cancer; and N/A: not applicable. Cells were left blank when this information was not reported in the article. ^a^ Time between prostate cancer diagnosis and brain metastasis diagnosis. ^b^ Prostate cancer type at time of brain metastasis diagnosis.

**Table 2 jcm-11-04165-t002:** Metastatic patterns.

Reference	% Parenchymal Metastasis	% Dural Metastasis	% Leptomeningeal Metastasis	% Bone Metastasis	% Nodal Metastasis	% Liver Metastasis	% Lung Metastasis
Myint ZW 2021 [4]				87		13	29
Boxley PJ 2021 [5]	31	69					
Bhambhvani HP 2020 [11]	100			100		35	48
Nguyen T 2020 [12]	100						
Ganau M 2020 [40]	29	71					
Ormond DR 2019 [46]	24	76		100			
Zhao F 2019 [44]				100			
Kanyılmaz G 2019 [13]	80	40		90	50	30	30
Hatzoglou V 2014 [38]	100			95	86		
Bartscht T 2014 [30]	100			50			
Gzell CE 2013 [39]	100			50	25	25	
Caffo O 2013 [7]	100						
Caffo O 2012 [47]	71		29				
Guedes B 2011 [37]		100					
Lawton A 2010 [45]		100		100	40	10	10
Flannery T 2010 [31]	60	90		60	30		
Lin C 2008 [10]			100	100			
Kim SH 2008 [32]	100			60			
Salvati M 2005 [9]	100			92		23	38
Tremont-Lukats IW 2003 [14]	100			95			
McCutcheon IE 1999 [34]	100			58	21	18	32
Nussbaum ER 1996 [36]	100						
Chung TS 1986 [42]	100						
Taylor HG 1984 [33]	43	86	7	100		71	36
Castaldo JE 1983 [43]	75	63	13	100		60	50
Sarma DP 1983 [35]	100			67	67	33	
Catane R 1976 [41]	100		40	100		100	100
Overall Median	N/A	N/A	N/A	95	40	28	34

N/A: not applicable. Cells were left blank when this information was not reported in the article.

**Table 3 jcm-11-04165-t003:** Brain metastases treatment.

Reference	Median Survival in Months (range) ^a^	Surgery (%)	Radiation (%)	Surgery and Radiation (%)	Supportive Care (%)	Other (%)	Type of Radiation Therapy
Myint ZW 2021 [4]	12	19				13	
Boxley PJ 2021 [5]							
Bhambhvani HP 2020 [11]	3 (0.4–25)		42	26	32		SRS
Nguyen T 2020 [12]	2		100				WBRT
Ganau M 2020 [40]				100			WBRT, SRS
Ormond DR 2019 [46]							
Zhao F 2019 [44]	10						
Kanyılmaz G 2019 [13]	4.5 (2–21)		100				WBRT
Hatzoglou V 2014 [38]	2.8						
Bartscht T 2014 [30]			100				WBRT
Gzell CE 2013 [39]	3.5 (2–24+)	25		75			WBRT
Caffo O 2013 [7]	2 (0.25–35.4)						
Caffo O 2012 [47]	4	3	32	16	48	35	WBRT, SRS
Guedes B 2011 [37]							
Lawton A 2010 [45]	6.17 (<1–15)		70		20	30	
Flannery T 2010 [31]	13		100				SRS
Lin C 2008 [10]			75		25		WBRT
Kim SH 2008 [32]	7 (6–22+)		80	20			WBRT, SRS
Salvati M 2005 [9]	13		23	77		8	WBRT, SRS
Tremont-Lukats IW 2003 [14]	1		24		76		SRS
McCutcheon IE 1999 [34]	4	3	76	21			WBRT
Nussbaum ER 1996 [36]	13						
Chung TS 1986 [42]	7.6 (mean)		71		29		WBRT
Taylor HG 1984 [33]							
Castaldo JE 1983 [43]	7		50		50		WBRT
Sarma DP 1983 [35]		25					
Catane R 1976 [41]			40				WBRT
# Studies Reporting	4.5 (median)	5	15	7	7	4	N/A

SRS: stereotactic radiosurgery and WBRT: whole-brain radiotherapy. Cells were left blank when this information was not reported in the article. ^a^ Median survival time from brain metastasis diagnosis (Salvati et al. and Castaldo et al. reported individual patient survival information, which was used to calculate a median survival of 13 and 7 months, respectively, by Kaplan–Meier analysis).

## Data Availability

Not applicable.

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
