# Peer review of "Prostate Cancer Brain Metastasis: Review of a Rare Complication with Limited Treatment Options and Poor Prognosis"

_jcm, 2022, doi:10.3390/jcm11144165_

Round 1
Reviewer 1 Report
Dear authors,
Congratulations for the review focused on a topic of not a primary importance in PC management, but potentially interesting in view of a ever greater survival due to availability of news agents.
I have no major complaints about this manuscript. This manuscript is well written and can help in a better understanding of prostate cancer metastases in the brain.
Moreover, the Figures and Tables were vividly presented, providing a variety of valuable information. Brain Metastases Treatment, and metastatic patterns in the tables they are well shown.
With the limitations of retrospective analysis the paper is interesting.
Author Response
Many thanks for the very positive feedback!
Reviewer 2 Report
This an important systematic review on brain metastases in metastatic prostate cancer, which is associated with significant morbidity, poor outcomes, and limited/unclear treatment options. While the evidence is limited, the authors have taken careful evaluation of the existing literature. The manuscript is well written.
Can the authors
1. include in the supplementary a more detailed search strategy?
2. clarify in Table 1 and section 3.6 - the Myint paper, not clear if every pt included was mCSPC vs mCRPC. Assuming most pts had mCRPC esp if median survival was 12 months.
3. include any available data on % pts with BM presenting with de novo metastatic disease vs recurrent at diagnosis?
4. Any data on whether some BM were hemorrahgic?
5. describe briefly what type of pts received WBRT vs SRS in more modern studies? # lesions/size/location etc. What type of systemic therapies did the pts received for BM?
6. what is the approximate incidence of BM in pts with non-adenohistology eg. neuroendocrine, ductal carcinoma, squamous; also HRD?
7. A few pts were long time survivers (>15 months). What were their disease/treatment characteristics if available?
8. comment on whether any specific subsets of pts should be screened for BM eg. neuroendocrine histology, extensive visceral disease, HRD? Detection rates of CT vs MRI?
9. comment on the ability of PARP inhibitors to penetrate BBB
10. Discuss some recent work in comparing molecular profiling of BM vs that in metastases outside of the brain, which suggests novel targeted therapeutic options?
11. Briefly mention if any trials are available including met prostate cancer with BMs, based on clinicaltrials.gov?
Author Response
We would like to thank Reviewer 2 for the overall very positive review and for the excellent points raised. The issues brought up have been addressed as follows:
- Include in the supplementary a more detailed search strategy?
- We applied a very broad search strategy using the search string and exclusion criteria detailed in the Methods section. Having said this we do not think that there is a need for further details in a supplementary section.
- Clarify in Table 1 and section 3.6 - the Myint paper, not clear if every pt included was mCSPC vs mCRPC. Assuming most pts had mCRPC esp if median survival was 12 months.
- We have contacted Dr. Myint to clarify these issues and have heard back, confirming that these are de novo metastatic prostate cancer patients.
- The median survival of patients with brain metastases described by Myint et al is around 12 months, i.e. longer than the median survival described in most publications on prostate cancer brain metastasis in patients with mCRPC.
- Include any available data on % pts with BM presenting with de novo metastatic disease vs recurrent at diagnosis?
- Information on patients with de novo vs recurrent BMs has been added in the Results section of the manuscript (lines 147-150). This type of information was rarely reported, yet the majority of informative studies reported a larger proportion of recurrent BM patients.
- Any data on whether some BM were hemorrhagic?
- Data on hemorrhagic BMs was only reported in 2 studies. This can be found outlined in the Results section (lines 172-174).
- Describe briefly what type of pts received WBRT vs SRS in more modern studies? # lesions/size/location etc. What type of systemic therapies did the pts received for BM?
- The characteristics of patients who received WBRT vs SRS were not reported in detail in any of the studies. Data on systemic therapies was reported in six studies and can be found in the Results section of the manuscript (lines 206-210).
- What is the approximate incidence of BM in pts with non-adeno histology eg. Neuroendocrine, ductal carcinoma, squamous; also HRD?
- Although an approximate incidence was not always reported, rare histological types of prostate cancer were found to have higher incidences of BMs. This data can be found in the Results section of the manuscript (lines 156-159).
- To the best of our knowledge the role of HRD in the risk of prostate cancer BM is not known. The recent analysis by Nguyen et al (Cell 2022 Feb 3;185(3):563-575.e11) did not reveal HDR defects as a driver of prostate cancer BM (a comment on the study by Nguyen et al has been added to the revised version of the manuscript, lines 296-300).
- A few pts were long time survivers (>15 months). What were their disease/treatment characteristics if available?
- Of the studies providing information on long-time survivors, two provided some disease and treatment characteristics of such patients, detailed in the Discussion section (line 305-309).
- Comment on whether any specific subsets of pts should be screened for BM eg. Neuroendocrine histology, extensive visceral disease, HRD? Detection rates of CT vs MRI?
- No solid, prospective information is available on specific types of patients or disease characteristics to guide screening for prostate cancer BMs. This is detailed in the Results section (lines 338-340).
- Comment on the ability of PARP inhibitors to penetrate BBB
- Most PARP inhibitors (niraparib is a notable exception) pass the blood brain barrier poorly. This can be found with detailed references in the Introduction section of the manuscript (lines 67-70).
- Discuss some recent work in comparing molecular profiling of BM vs that in metastases outside of the brain, which suggests novel targeted therapeutic options?
- The revised version of the manuscript contains additional ‘molecular’ information that can be found in the Discussion section (lines 296-300).
- Briefly mention if any trials are available including met prostate cancer with BMs, based on clinicaltrials.gov?
- We were not able to identify clinical trials focusing specifically on prostate cancer BM, but prostate cancer patients are eligible for certain BM trials. This has been detailed in the Discussion section (lines 347-351).